# ZERO-SHOT OBJECT-LEVEL OUT-OF-DISTRIBUTION DETECTION WITH CONTEXT-AWARE INPAINTING

## ABSTRACT

Detecting when an object detector predicts wrongly, for example, misrecognizing an out-of-distribution (ODD) unseen object as a seen one, is crucial to ensure the model's trustworthiness. Modern object detectors are known to be overly confident, making it hard to rely solely on their responses to detect error cases. We therefore investigate the use of an auxiliary model for the rescue. Specifically, we leverage an off-the-shelf text-to-image generative model (e.g., Stable Diffusion), whose training objective is different from discriminative models. We surmise such a discrepancy would allow us to use their inconsistency as an error indicator. Concretely, given a detected object box and the predicted class label, we perform class-conditioned inpainting on the box-removed image. When the predicted object label is incorrect, the inpainted image is doomed to deviate from the original one, making the reconstruction error an effective recognition error indicator, especially on misclassified OOD samples. Extensive experiments demonstrate that our approach consistently outperforms prior zero-shot and non-zero-shot OOD detection approaches.

## 1 INTRODUCTION

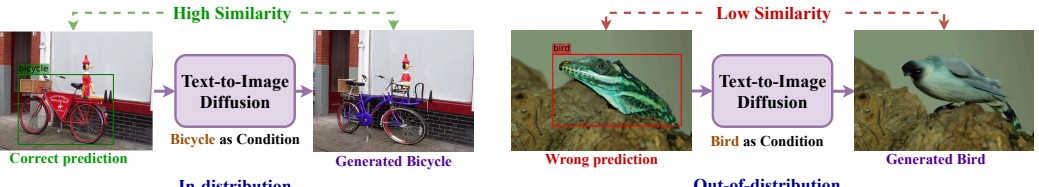

Figure 1: **Intuition behind RONIN for zero-shot OOD object detection**. An object detection model predicts bounding boxes and class labels on the input image, making correct classifications on ID objects and misclassifications on OOD objects. RONIN leverages that outcome for inpainting the predicted objects through a strong text-to-image diffusion model. The ID objects are then inpainted similarly, while the OOD objects are inpainted differently. A feature-based alignment assessment can reflect this difference, therefore detecting OOD.

Object detection systems have made significant advancements in recent years, and are widely applied in our lives both in online and offline settings, including environmental science (Beery et al., 2019), manufacturing (Ahmad & Rahimi, 2022), or even healthcare (Ragab et al., 2024). It is thus crucial to ensure the reliability and robustness of these systems. However, one typical error is that the detector sometimes misidentifies something as an object from its predefined classes. While detector confidence is commonly used for filtering out erroneous detections, overconfidence has been observed across many architectures and domains (Pathiraja et al., 2023). This highlights the challenge of relying solely on the internal responses of an object detector's to identify these errors.

Various existing works address this challenge during training. Munir et al. (2022); Pathiraja et al. (2023) introduce auxiliary losses to regularize confidence calibration. Du et al. (2022b;a) approach the problem as an out-of-distribution (OOD) detection task and learn better detector features that are more compatible with classic OOD detection methods. Although these methods are effective, many practitioners use pre-trained, publicly available models for their applications and are unable to retrain the detectors to incorporate these techniques.

In this paper, we explore the use of additional pre-trained models as auxiliary tools to address these challenges. In particular, recent text-conditioned generative models (Rombach et al., 2021; Saharia et al., 2022) demonstrate strong language understanding and image synthesis capabilities and show potential for augmenting various discriminative tasks (Li et al., 2023a). We build upon such insights and leverage these models as an additional source of information to help identify erroneous detections.

To this end, we propose ZeRo-shot OOD CONtextual INpainting (**RONIN**), as illustrated in Figure 1. RONIN operates upon detected bounding boxes and predicted class labels from a detector. For each class in the detector's vocabulary, RONIN masks out all detected instances in an image and performs inpainting conditioned on the class name. It then compares the similarities between the original detected objects and their inpainted counterparts, as well as both objects' similarities to the class labels. Typically, erroneous detections which by definition are incompatible with class labels, will look dissimilar from the inpainted objects. Correct detections, however, will share similar semantics with their inpainted ones. This makes OOD objects easy to distinguish. We evaluate RONIN on real-world benchmark datasets and show that it consistently outperforms existing methods. We also provide in-depth analysis and visualizations to further validate the effectiveness of RONIN.

**Remark.** In this paper, we mainly focus on the error cases of object detection when wrongly recognizing unseen OOD objects as seen ID ones, i.e., object-level OOD detection. However, with the large-scale off-the-shelf generative model, RONIN is general enough to also perform the correction when encountering a wrong prediction on a seen ID object, such as "*cat instead of dog*". While in our experiment we mainly explore the OOD detection task because it is considered the harder case, we conducted two preliminary studies on: (i) **ID misclassification** (Section 6.1), where the inpainting process helps RONIN to correct the false prediction encountered on ID samples; and (ii) ***near*-OOD detection** (Section 6.2), where pre-anticipate closest ID labels as specific inpainting condition support RONIN to tackle near-OOD problems. We hope that this study opens the potential for future studies on both OOD detection and misclassification from different perspectives.

## 2 RELATED WORK

**Out-of-distribution Detection.** The goal of the out-of-distribution (OOD) detection task is to determine whether a given sample belongs to a certain distribution. It has been widely studied at the *image level*, where a whole image is treated as a sample. Common approaches include leveraging classifier-specific information such as confidence scores (Hendrycks & Gimpel, 2016; Lee et al., 2017; Liang et al., 2018; DeVries & Taylor, 2018; Hsu et al., 2020; Liu et al., 2020; Wei et al., 2022) or learned features (Lee et al., 2018; Denouden et al., 2018; Tack et al., 2020; Sehwag et al., 2021; Xiao et al., 2021; Sun et al., 2022), or directly modeling an image distribution using generative models (Ren et al., 2019; Serrà et al., 2019; Xiao et al., 2020; Schlegl et al., 2017; Zong et al., 2018; Graham et al., 2023; Liu et al., 2023; Li et al., 2023b). Recent works (Ming et al., 2022; Esmaeilpour et al., 2022; Wang et al., 2023) have also explored using CLIP (Radford et al., 2021) to identify OOD examples in a zero-shot manner, bypassing the need to learn from in-domain data explicitly.

OOD detection can also be extended to the *object-level*, where objects within an image are treated as individual samples. Most existing research in this setting focuses on training-time interventions. For instance, Du et al. (2022b;a) improve detector features to make them more separable between in-distribution (ID) and OOD data; Wilson et al. (2023) uses adversarial examples to train an MLP for classifying ID and OOD instances. In contrast, our approach RONIN explores the object-level setting through post-hoc interventions instead.

**Text-to-Image Generative Models.** Recent advances in generative modeling have made large-scale text-to-image models widely available (Rombach et al., 2021; Ramesh et al., 2022; Saharia et al., 2022). These models exhibit a deep understanding of language and are highly effective at generating or editing high-quality images from diverse prompts, offering a promising approach to data synthesis across various tasks. Furthermore, recent works have shown that these models can also enhance discriminative tasks. For instance, Li et al. (2023a); Jaini et al. (2024) demonstrate that Stable Diffusion (Rombach et al., 2021) can function as an effective zero-shot classifier, achieving accuracy comparable to or surpassing CLIP and various trained discriminative classifiers. There has also been recent progress in applying text-conditioned diffusion models for image-level OOD detection (Du

et al., 2023; Gao et al., 2023; Fang et al., 2024). Unlike these works, RONIN focuses on exploring the use of such models for object-level tasks.

## 3 PROBLEM FORMULATION

We address the task of object-level out-of-distribution (OOD) detection. Specifically, given an object detector trained to detect a pre-defined set of categories (e.g., different kinds of vehicles), we aim to identify the error cases when it wrongly recognizes a novel object (e.g., a wild animal) as one of the pre-defined categories and detects it. We refer to the pre-defined set of categories as the in-distribution (ID) classes, following Du et al. (2022b); the novel categories as the OOD classes.

Formally, given an image $\mathbf{x}$ and an object detector $f(.; \theta)$ trained for the ID classes $\mathcal{Y}^{in}$, $f(\mathbf{x}; \theta)$ outputs a list of bounding boxes $\boldsymbol{b} = \{b_1, b_2, \ldots, b_n\}$ and their associated ID class labels $\hat{\boldsymbol{y}} = \{\hat{y}_1, \hat{y}_2, \ldots, \hat{y}_n\}$, where $\hat{y}_i \in \mathcal{Y}^{in}$. Object-level OOD detection is then formulated as a binary classification problem, classifying whether the object within $b_i$ truly belongs to ID classes. Typically, one would develop a scoring function $\boldsymbol{g}$ such that given a bounding box and its predicted label $(b, \hat{y})$, $\boldsymbol{g}$ gives a higher score $\boldsymbol{g}(b, \hat{y})$ if $b$ outlines an ID object and a lower score if it outlines an OOD object.

Existing works (Du et al., 2022b;a) proposed a specific training process for the object detector so that its responses (e.g., feature vector of each bounding box) would better distinguish between ID and OOD classes. However, doing so requires re-training the object detector, assuming access to the original training data and making it infeasible to off-the-shelf object detectors.

**Aim.** To address this limitation, we aim to design an OOD detection mechanism that is **post-hoc**, without the need to modify (e.g., fine-tune) the pre-trained object detector, and **zero-shot** (Ming et al., 2022), without the need to access the original training data or any ID-class data. This sharply contrasts several prior post-hoc methods that need ID-class data (Ren et al., 2019; Xiao et al., 2020).

**Approach.** The emergence of vision-language foundation models (Radford et al., 2021; Li et al., 2022; Liu et al., 2024a) trained on abundant image and free-form text pairs has gradually removed the boundary between the closed-set and open-set settings Li et al. (2023a). For example, while not perfect, CLIP (Radford et al., 2021) can match an image with unbounded concepts. Such a zero-shot capability has been leveraged in prior work to detect OOD samples given a set of ID concepts (Ming et al., 2022; Esmaeilpour et al., 2022; Wang et al., 2023). In this work, we leverage another kind of foundation model, text-to-image generative models (Rombach et al., 2021), capable of generating images given free-form texts. While not designed for object detection and optimized for the ID data, we surmise their built-in, generic capability would facilitate object-level OOD detection.

## 4 RONIN: ZERO-SHOT OOD CONTEXT INPAINTING

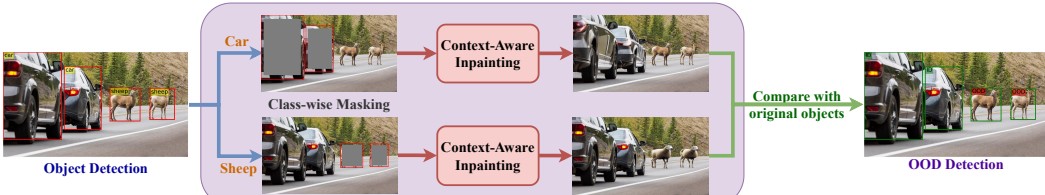

Figure 2: **Overall framework of RONIN** with three main components: pre-trained object detection, context-aware inpainting, and vision-language similarity assessment. Given an input image, the ID-trained object detector makes correct predictions (*car*) on ID and incorrect predictions (*sheep*) on OOD objects. With diffusion models, RONIN synthesize realistic objects with the predicted labels as conditions. Finally, the similarities between the original detected objects and their inpainted counterparts in multiple ways are then utilized for OOD detection.

We provide a high-level overview of our method RONIN in Figure 2 for object-level OOD detection. With bounding boxes and label predictions from the object detection model, RONIN leverages inpainting to generate context-aware in-distribution (ID) objects. The similarities between inpainted

ID objects and the original objects are computed, and the higher the similarity, the more likely the original objects are ID. Below, we explain individual components in detail.

## 4.1 CONTEXT-AWARE INPAINTING

Our key idea to propose RONIN is to leverage diffusion models to synthesize in-distribution objects through inpainting, lifting them closer to the in-distribution domain. Specifically, a black box object prediction $f(.|\theta)$ trained on $\mathcal{Y}^{in}$ performs the prediction a $(\boldsymbol{b}, \hat{y})$ on input image $\mathbf{x}_{ori}$, where $\boldsymbol{b}$ are the coordinates of the bounding box and $\hat{y} \in \mathcal{Y}^{in}$ is the predicted label. The side ratio $r$ defines the relationship between the mask $m$ and the bounding box $\boldsymbol{b}$. Specifically, the ratio of the area covered by $\boldsymbol{m}$ to the area of the bounding box $\boldsymbol{b}$ is given by $\frac{|\boldsymbol{m}|}{|\boldsymbol{b}|} = r^2$. Rather than considering the entire image, we constrain the background $s$ as the region between $\boldsymbol{b}$ and $\boldsymbol{m}$, i.e., $s$ is the area outside $\boldsymbol{m}$ but within $\boldsymbol{b}$. Since the $\boldsymbol{b}$ fitly covers the input object while $\boldsymbol{m}$ is a smaller rectangle placed in the center of it, $s$ opportunistically retains sufficient *"semantic"* information about the original object, including shape, color, and visual appearance, for effective synthesizing.

RONIN factors in the foreground mask $\boldsymbol{m}$, the predicted label $\boldsymbol{c} = \hat{y}$, and the semantic background $s$ as the conditions for inpainting contextually. In particular, given $\mathbf{x}_T = \mathbf{x}_{ori}$, RONIN utilized a pre-trained text and shape-guided Denoising Diffusion Probabilistic Models (DDPM) Ho et al. (2020) to generate a random Gaussian noise over $\boldsymbol{m}$ and perform the reverse diffusion process with $T$ steps, guided by $\boldsymbol{c}$, to obtain the inpainting $\mathbf{x}_{inp} = \mathbf{x}_0$, as in Equation 1, where $\mu_\theta$ and $\Sigma_\theta$ are priorly learned by training a neural network with parameter $\theta$.

$$p_\theta(\tilde{\mathbf{x}}_{t-1}|\mathbf{x}_t, \, \boldsymbol{c}) = \mathcal{N}(\tilde{\mathbf{x}}_{t-1}; \, \mu_\theta(\mathbf{x}_t, \, t, \, \boldsymbol{c}), \Sigma_\theta(\mathbf{x}_t, \, t, \, \boldsymbol{c})) \tag{1}$$

$$\mathbf{x}_{t-1} = \tilde{\mathbf{x}}_{t-1} \odot \boldsymbol{m} + \boldsymbol{s} \tag{2}$$

While inpainted objects are synthesized based on $\hat{y}$ over the foreground region $\boldsymbol{m}$, information retained from $s$ guides DDPM to retain the overall appearance of the original object, as in Equation 2. Therefore, the inpainted outcome will have the same visual and semantic characteristics as the original object, despite being presented differently. This supports RONIN to synthesize objects closely similar to ID original ones and largely different in OOD cases, as illustrated in Figure 2. We further analyze the impact of $\boldsymbol{m}$ and $s$ on the performance of RONIN in Section 5.3 and Appendix D.

## 4.2 OBJECT-WISE VS. CLASS-WISE INPAINTING

For the masking and inpainting process, two natural approaches can be considered: the *"object-wise"* strategy and the *"class-wise"* strategy. The object-wise strategy involves inpainting each object independently. This approach allows for detailed synthesis, potentially leading to more accurate recognition of out-of-distribution (OOD) objects because each object's unique features are preserved and refined individually. Consequently, this can enhance the overall performance of the OOD object detection task, especially in scenarios where detailed object reconstruction is crucial.

On the other hand, the class-wise strategy involves inpainting all objects that share the same predicted class label by taking the union of all bounding boxes $\boldsymbol{b}_i$ with the same $\hat{y}$ and performing inpainting for the entire class at once. This approach is significantly more efficient, particularly in images containing numerous small and distinct objects of the same class, such as multiple cars in an urban street scene. The efficiency gains are due to the batch-inpainting nature of this method, where the computational cost of inpainting scales with $|\mathcal{Y}^{in}|$— the number of distinct classes — rather than the total number of individual objects.

In our ablation study in Section 5.3, we analyze the performance of both strategies under various settings. While the object-wise strategy provides a higher level of detail and accuracy in recognizing OOD objects due to its individualized approach, the class-wise strategy offers substantial efficiency improvements, making it more practical for scenes with many objects belonging to the same class. This trade-off between accuracy and efficiency is critical in determining the appropriate inpainting strategy based on the specific requirements and constraints of the application at hand.

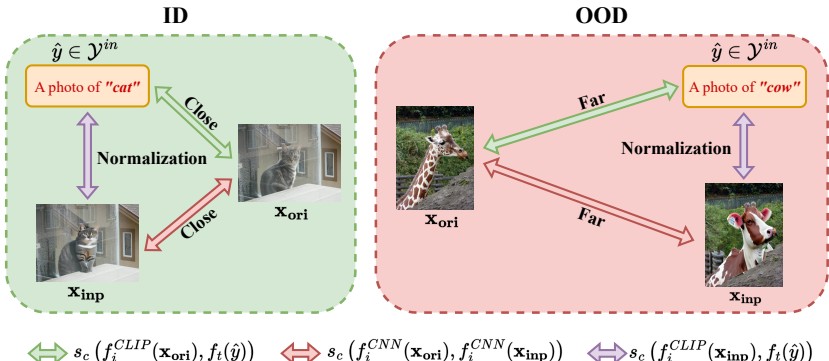

**Figure 3: Triplet similarity relations** between the original object $\mathbf{x_{ori}}$ and inpainted object $\mathbf{x_{inp}}$ through the predicted class label $\hat{y}$. ID objects are more likely to have higher similarity between $\mathbf{x_{ori}}$ and $\hat{y}$, and between $\mathbf{x_{ori}}$ and $\mathbf{x_{inp}}$ than OOD objects.

## 4.3 VISION-LANGUAGE SIMILARITY ASSESSMENT

After performing the inpainting, we end up with the original object, denoted as $\mathbf{x_{ori}}$, and the inpainted object, denoted as $\mathbf{x_{inp}}$. In cases where the object is ID, the similarity between $\mathbf{x_{ori}}$ and $\mathbf{x_{inp}}$ tends to be higher, indicating that the inpainting process preserves the essential features of the original object. Conversely, with OOD objects, this similarity is expected to be lower, suggesting that the inpainted object diverges significantly from the original.

However, the degree of similarity between $\mathbf{x_{ori}}$ and $\mathbf{x_{inp}}$ can vary significantly due to the stochastic nature of the diffusion inpainting process. To address this variability and enhance the distinction between ID and OOD cases, we introduce the predicted class label $\hat{y} \in \mathcal{Y}^{in}$ to form a triplet of similarity scores, allowing for a more nuanced assessment between $\mathbf{x_{ori}}$ and $\mathbf{x_{inp}}$:

$$
\begin{aligned}
similarity_{(ori,\ \hat{y})} &= s_c\left(f_i^{CLIP}(\mathbf{x_{ori}}), f_t(\hat{y})\right) \\
similarity_{(inp,\ \hat{y})} &= s_c\left(f_i^{CLIP}(\mathbf{x_{inp}}), f_t(\hat{y})\right) \\
similarity_{(ori,\ inp)} &= s_c\left(f_i^{CNN}(\mathbf{x_{ori}}), f_i^{CNN}(\mathbf{x_{inp}})\right)
\end{aligned}
\tag{3}
$$

Where $f_i^{CLIP}$ and $f_t$ are visual and textual representations from vision-language correlation model CLIP (Radford et al., 2021), $f_i^{CNN}$ are visual representations from visual feature extraction SimCLRv2 (Chen et al., 2020), and $s_c(a,\ b)$ for Cosine similarity. Interestingly, high $similarity_{(ori,\ inp)}$ favors in-distribution object, while $similarity_{(inp,\ \hat{y})}$ can operate as a normalizing term concerning $\hat{y}$ since some labels naturally have high or low similarity between $\mathbf{x_{ori}}$. We present Figure 3 to visualize the relationship between all three similarities in both ID and OOD cases. Based on that, we then form a triplet OOD score:

$$
S_{\text{triplet}} = \frac{similarity_{(ori,\ \hat{y})}^{\alpha} \times similarity_{(ori,\ inp)}^{\beta}}{similarity_{(inp,\ \hat{y})}}
\tag{4}
$$

where $\alpha = 1$ and $\beta = 2$ are exponential hyperparameters to either favor the visual-language $similarity_{(ori,\ \hat{y})}$ or the visual $similarity_{(ori,\ \hat{y})}$. Based on the developed intuitions, higher scores suggest a higher likelihood of the input object being in-distribution (ID), and vice versa. In practice, one can use a thresholding method to identify OOD objects effectively.

## 5 EXPERIMENTS

### 5.1 EXPERIMENTAL SETUPS

**Datasets.** We use Pascal-VOC (Everingham et al., 2010) and Berkerly DeepDrive (BDD-100k) (Yu et al., 2020) as ID data, and MS-COCO (Lin et al., 2014) and OpenImages (Kuznetsova et al., 2020) as OOD data. We follow the splits from Du et al. (2022a;b), where *overlapping categories in the ID and OOD datasets are removed*. We select a subset of 200 images from the test set of BDD-100k and 400 each from the test sets of Pascal-VOC, MS-COCO, and OpenImages. Data processing details can be found in Appendix A.

**Evaluation Metrics.** We evaluate the OOD detection performance using the area under the Receiver Operating Characteristic curve (**AUROC**) and the false positive rate of OOD objects at 95% true positive rate of in-distribution objects (**FPR@95**). We additionally report the mean Average Precision (**mAP**) with IoU=0.5:0.95 on the in-distribution data to validate the quality of the object detectors.

**Baselines.** We consider two categories of zero-shot baselines: (1) *Discriminative approaches*: **MCM** (Ming et al., 2022), CLIP-based **ODIN** (Liang et al., 2018), CLIP-based **Energy Score** (Liu et al., 2020), and CLIPN (Wang et al., 2023). (2) *Alternative generative approaches*: first generating synthetic data and then applying standard OOD detection methods such as **Mahalanobis** (Xiao et al., 2021) and **KNN** (Sun et al., 2022). Additionally, for a comprehensive comparison, we also include two representative training-based object-level methods **VOS** (Du et al., 2022b) and **SIREN** (Du et al., 2022a), resulting in a total of **seven** baselines. Implementation details can be found in Appendix B.

**Implementation Details.** For RONIN, we perform class-wise inpainting with Stable Diffusion 2 Inpainting (Rombach et al., 2021) with 20 steps. Each object is masked with a center mask covering 0.9 of the height and width of its original bounding box. Object-label similarities are determined by the distance between OpenCLIP (Ilharco et al., 2021) features, and object-object similarities by the SimCLRv2 (Chen et al., 2020) features. The triplet OOD score is calculated with $\alpha = 2$ and $\beta = 1$. For both RONIN and the baselines, the detected objects are obtained from the Deformable-DETR (Zhu et al., 2020) detectors from Du et al. (2022a) to ensure comparability.

### 5.2 EXPERIMENTAL RESULTS

Table 1: **Main results**. Comparison of RONIN with competitive OOD detection baselines, with Deformerable DETR as object detector. All results are reported in percentage. RONIN demonstrates strong and consistent performance across settings.

| ID dataset | mAP (ID) | Method | MS-COCO | | OpenImages | |
|---|---|---|---|---|---|---|
| | | | FPR@95 ($\downarrow$) | AUROC ($\uparrow$) | FPR@95 ($\downarrow$) | AUROC ($\uparrow$) |
| **PASCAL-VOC** | 60.89 | ODIN (Liang et al., 2018) | 41.65 | 88.22 | 55.87 | 86.46 |
| | | MCM (Ming et al., 2022) | 62.47 | 83.15 | 71.52 | 81.45 |
| | | Energy Score (Liu et al., 2020) | 29.48 | 90.26 | 24.57 | 91.24 |
| | | Mahalanobis (Xiao et al., 2021) | 63.30 | 83.26 | 45.22 | 87.11 |
| | | KNN (Sun et al., 2022) | 58.56 | 85.52 | 45.00 | 84.62 |
| | | VOS (Du et al., 2022b) | 48.15 | 88.75 | 54.63 | 83.65 |
| | | SIREN (KNN) (Du et al., 2022a) | 64.70 | 78.68 | 66.69 | 75.12 |
| | | CLIPN (Wang et al., 2023) | 43.09 | 85.45 | 41.74 | 89.31 |
| | | **RONIN** | 27.42 | 91.09 | 18.04 | 93.34 |
| **BDD-100k** | 31.30 | ODIN | 96.51 | 55.18 | 95.56 | 57.87 |
| | | MCM | 95.56 | 55.82 | 92.22 | 57.05 |
| | | Energy Score | 71.75 | 74.49 | 53.33 | 73.09 |
| | | Mahalanobis | 31.75 | 90.74 | 23.33 | 93.81 |
| | | KNN | 35.87 | 86.58 | 31.11 | 92.75 |
| | | VOS | 65.45 | 78.34 | 59.23 | 80.42 |
| | | SIREN (KNN) | 42.86 | 89.37 | 37.97 | 91.78 |
| | | CLIPN | 28.49 | 92.14 | 44.76 | 85.78 |
| | | **RONIN** | 30.16 | 92.77 | 30.00 | 91.60 |

Table 1 presents the performance of RONIN and the baselines based on the DeformableDETR detections in four settings. RONIN achieves consistent and competitive performance across all four

OOD scenarios and attains the highest performance in three of them. Notably, in PascalVOC vs. OpenImages, it improves FPR@95 by 23% over the best-performing baseline.

Furthermore, RONIN outperforms the training-based methods VOS and SIREN in three settings and matches their performance in one. This demonstrates the effectiveness of using off-the-shelf models to bypass the need for retraining with dedicated in-domain data. Purely CLIP-based zero-shot approaches show fluctuating results, while alternative generative approaches combined with standard OOD detection perform competitively despite still underperforming RONIN. This further highlights the advantages of incorporating generative models for object-level OOD detection.

We further report the performance of RONIN and the baselines with Faster R-CNN (Ren et al., 2016) object detector in Appendix C. RONIN continues to outperform the baselines in this setting.

## 5.3 ABLATION STUDIES

Table 2: Ablation on the rescaling ratio between foreground masking and semantic background.

| Covered masking | **PascalVOC - MS COCO** | |
| ratio $m$ | FPR@95 ($\downarrow$) | AUROC ($\uparrow$) |
|---|---|---|
| 25% | 42.89 | 89.17 |
| 56% | 37.32 | 90.01 |
| 81% | 27.42 | 91.09 |
| 90% | 28.66 | 90.99 |
| 100% | 31.13 | 90.01 |

**Inpainting masking ratio.** Table 2 presents the trade-off between the sizes of foreground masking $m$ and semantic background $s$, toward context-aware inpainting. As described in Section 4.1, the mask $m$ specifies the regions where the inpainting occurs, while the semantic background $s$ preserves information from the original object for contextual inpainting. A larger $s$ resulting in a small $m$ inpainting region is ineffective in differentiating inpainted and original objects in the OOD cases. In contrast, too small $m$ with limited visual context might result in a distinct inpainted object in both ID and OOD cases, leading to suboptimal performances. Empirical results show that optimal performance is achieved when the mask $m$ covers approximately 80% of the bounding box, leaving about 20% of the predicted region for semantic background $s$. With this ratio, RONIN retains enough information in $s$ to generate similar objects for ID cases and allows sufficient $m$ to synthesize dissimilar objects for OOD cases. Additional visualizations and analysis can be found in Appendix D.

Table 3: OOD detection performances between object-wise and class-wise inpainting.

| | **FPR@95** ($\downarrow$) | | **AUROC** ($\uparrow$) | |
| | Object-wise | Class-wise | Object-wise | Class-wise |
|---|---|---|---|---|
| VOC - COCO | 29.48 | 27.42 | 90.94 | 91.09 |
| VOC - OpenImages | 20.00 | 18.91 | 93.13 | 92.69 |
| BDD - COCO | 26.35 | 30.16 | 93.47 | 92.77 |
| BDD - OpenImages | 24.44 | 30.00 | 91.92 | 91.60 |

**Class-wise vs. object-wise inpainting.** As an alternative to masking and inpainting all objects of the same predicted class, one can choose to mask and inpaint each object individually. We name this strategy object-wise inpainting, and compare its performance in Table 3. Results show that the two strategies yield comparable performance, but the object-wise approach is 1.5-3 times slower than the class-wise approach, depending on the object density in a frame. Therefore, class-wise inpainting is generally preferred. More details can be found in Appendix E.

**Abblations on triplet measurement design.** In Table 4 presented two ablation studies on validating the design choices of the triplet similarity (Equation 4). The first study (*lines 1–4*) highlights the importance of incorporating all three pairing similarities, which is crucial for consistent and effective measurement between original $\mathbf{x_{ori}}$ and inpainted $\mathbf{x_{inp}}$, given the predicted ID label $\hat{y}$. The second

study (*lines 5–7*) demonstrates that prioritizing the rich vision-language similarity, $similarity(ori, \hat{y})$, over the visual-based similarity, $similarity(inp, ori)$, i.e., setting $\alpha > \beta$, enables RONIN to achieve peak performance. This finding aligns with a similar ablation study conducted by Ming et al. (2022). Additionally, adjusting the values of $\alpha$ and $\beta$ within this insight can lead to performance differences, allowing RONIN to be flexibly adapted for specific practical needs. However, our experiments suggest that maintaining $\beta = 1$ is the most stable choice across settings, and increasing $\alpha > 4$ yields diminishing impacts. For simplicity and consistency, we set $\beta = 1$ and $\alpha = 2$ in our triplet similarity design, as defined in Equation 4.

Table 4: Ablation studies on the design of the triplet measurement (Equation 4) reveal that (i) all three similarities are essential for providing a stable and effective triplet score, and (ii) prioritizing the vision-language similarity enables RONIN to achieve peak performance.

|  | COCO / OpenImages | |
|---|---|---|
|  | FPR@95 ($\downarrow$) | AUROC ($\uparrow$) |
| without $similarity(inp, \hat{y})$ (no normalization) | 44.54 / 37.39 | 89.77 / 91.42 |
| without $similarity(ori, \hat{y})$ ($\alpha = 0$; $\beta = 1$) | 65.15 / 68.91 | 69.72 / 7 8.53 |
| without $similarity(ori, inp)$ ($\alpha = 1$; $\beta = 0$) | 30.93 / 36.96 | 86.14 / 90.10 |
| all three similarities (triplet score, $\alpha = \beta = 1$) | 35.46 / 24.78 | 87.75 / 91.43 |
| triplet similarity, $\alpha = 1$; $\beta = 2$ | 43.93 / 33.91 | 85.40 / 89.50 |
| triplet similarity, $\alpha = 2$; $\beta = 1$ | 27.42 / 18.04 | 91.41 / 93.34 |
| triplet similarity, $\alpha = 2$; $\beta = 2$ | 35.46 / 25.43 | 90.32 / 95.19 |

**Choice of feature representations.** In Table 5, we examine the impact of using a different feature source by replacing SimCLRv2 features with the OpenCLIP image encoder for $similarity_{(ori, inp)}$. RONIN with OpenCLIP image features achieves comparable performance, demonstrating its robustness to the choice of feature representations.

Table 5: RONIN performances with alternative image feature choice on PascalVOC-as-ID settings.

|  | COCO / OpenImages | |
|---|---|---|
|  | FPR@95 ($\downarrow$) | AUROC ($\uparrow$) |
| RONIN + SimCLRv2 image features | 27.42 / 18.04 | 91.09 / 93.34 |
| RONIN + OpenCLIP image features | 37.73 / 28.26 | 89.14 / 91.08 |

### 5.4 QUALITATIVE ANALYSIS

**Success Cases Analysis.** Figure 4 showcases the performances of RONIN in the PascalVOC as ID settings. In summary, RONIN effectively differentiates between ID and OOD by closely replicating the original objects in ID inpainting, resulting in high similarity scores, while creating distinct differences in OOD inpainting, which yields low scores. RONIN also performed coherently even with cluttered or occluded objects, such as a group of cyclists or elephants, or even drawings, showing the effectiveness of our approach. We provide more visualizations in Appendix F.

**Failure Cases Analysis.** Under certain challenging scenarios, RONIN may perform suboptimally. For instance, when the predicted box is too small, the inpainting model may fail to synthesize the object, leading to low similarities for ID cases. This can be mitigated by enhanced training of the inpainting model (Wei et al., 2021; Ouyang et al., 2022). Additionally, when OOD objects highly resemble ID categories both visually and semantically (e.g. "*zebra vs. horse*"), the inpainting model sometimes produces higher similarity reconstructions for OOD objects. This can potentially be alleviated with more precise prompting in the following sessions.

## 6 PRELIMINARY STUDIES

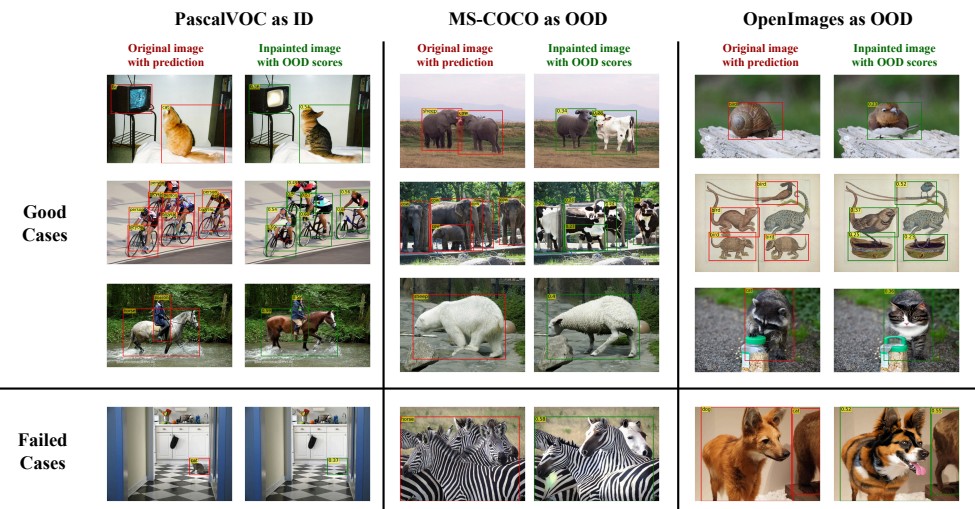

Figure 4: **Qualitative visualization of RONIN** on PascalVOC as ID and MS-COCO/OpenImages as OOD. The left and right columns are original images with ID predictions and contextually inpainted objects with OOD scores. We present three examples where RONIN failed to perform the correct OOD detection due to the limitation of the diffusion models (the figure is best viewed in color).

In this paper, we primarily focus on addressing the misrecognition of unseen objects (OOD) as seen ones, known as object-level out-of-distribution detection. However, the general knowledge provided by the generative model enables RONIN to (i) correct errors in the detection model and (ii) identify *near*-OOD objects where the unseen objects strike very strong resemblance to the seen ones, which is actually a very challenging task in OOD detection. We conducted two preliminary studies to investigate how RONIN can tackle these two problems.

## 6.1 RONIN FOR MISCLASSIFIED IN-DISTRIBUTION DETECTION

In the first study, we investigate how RONIN also corrects errors in the in-distribution (ID) data when the object detection model makes a misclassification. The intuition behind this is that if a misclassification occurs, RONIN can perform similarly well as it does in OOD detection. This is because misclassified objects will also have low triplet scores, akin to OOD objects. Since misclassifications are rare in our experiments, we simulate these scenarios by replacing the correct label, i.e., the label with the highest score from the detection model, with the second-highest score label. For instance, in PascalVOC (ID data), typical simulated misclassifications include cases like *"cat vs. dog"* or *"motorcycle vs. bicycle"*, which exhibit both visual and semantic similarities.

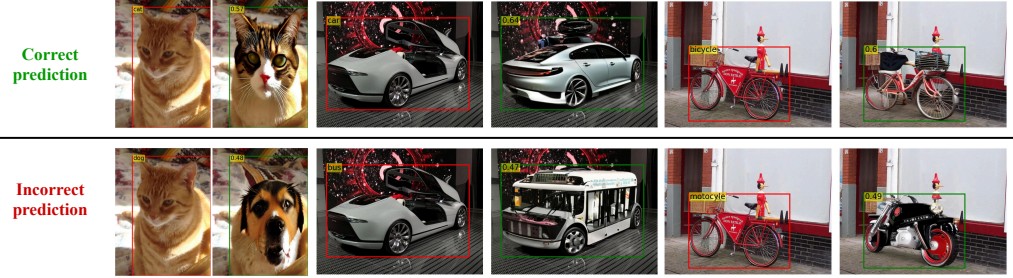

Figure 5: **Examples of misclassified ID recognition** on in-distribution PascalVOC samples. Given a prediction either with the correct or incorrect classified labels, RONIN can distinguish them by assigning low triplet scores to misclassified objects and high scores to correct ones.

Figure 5 provides qualitative comparisons between correct and incorrect label predictions using RONIN 's triplet similarity score. RONIN effectively assigns low scores to misclassified ID objects while favoring higher scores for correct predictions. We believe that this preliminary experiment paves

the way for more in-depth investigations into the potential of generative models in both zero-shot OOD detection and ID misclassification detection.

## 6.2 RONIN FOR NEAR-OOD DETECTION

The robustness of off-the-shelf generative models can support RONIN to effectively handle near-OOD detection, a challenging task for standard OOD detection algorithms, by leveraging a strong discriminative model to control the generative process based on specific instructions. For instance, consider an input photo containing an OOD object, such as a "*zebra*," misclassified as the ID label "*horse*" (Figure 6, left column). Due to the similarity between these classes, directly instructing RONIN to inpaint a horse might result in a modified image that closely resembles the original zebra, rendering near-OOD detection ineffective (Figure 6, middle column). To address this, we pre-anticipate near-OOD classes for the given ID label as exclusions. For example, if the ID class is "*horse*," we can roughly identify near-OOD classes like "*zebra*" or "*donkey*" and condition the diffusion model to generate a "*horse*" that distinctly differs from these near-OOD objects (Figure 6, right column). This approach ensures inpainted results better align with the predicted ID label, enhancing the robustness of our framework to near-OOD scenarios.

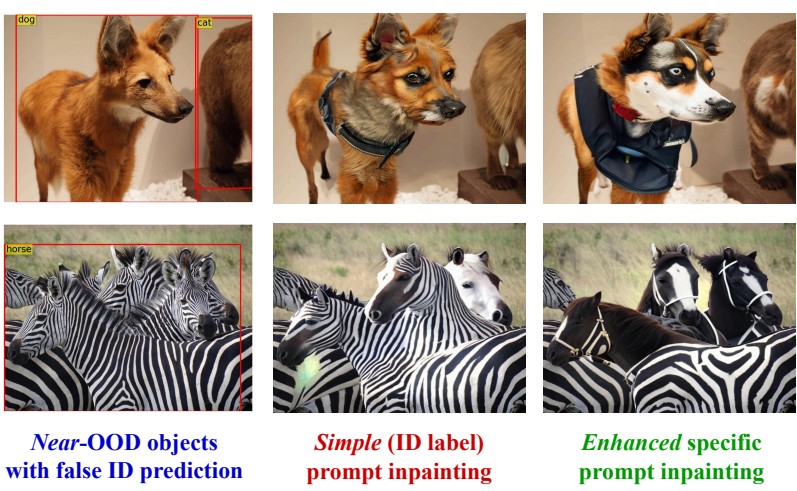

**Near-OOD objects**
**with false ID prediction**
    **Simple (ID label)**
**prompt inpainting**
    **Enhanced specific**
**prompt inpainting**

Figure 6: **Inpainting performance on hard OOD cases** with the conventional simple prompt (ID label only) and the enhanced specific prompt (ID label and closest related species as supplementary exclusions) for *near*-OOD detection.

## 7 DISCUSSION

Overall, RONIN demonstrates strong performance in identifying erroneous detections across diverse scenarios. Even with open-vocabulary object detectors, which have the ability to recognize any kind of object given a specific prompt, close-set definitions of the target objects are still required, allowing RONIN to identify out-of-vocabulary objects or potentially correct the erroneous predictions flexibly. Admittedly, since current leading text-conditioned inpainting models are diffusion-based, this creates challenges in using RONIN in applications where real-time detection is critical. While low denoising-step schedules can accelerate the time performance, they often compromise both the inpainting quality and OOD detection performances of RONIN. Nevertheless, for those domains, RONIN remains valuable for retrospectively analyzing the failure modes of deployed models. The advancements in diffusion model acceleration (Song et al., 2023; Liu et al., 2024b) and the development of inpainting models based on other approaches (e.g. GAN) also show great promise for enabling RONIN in real-time scenarios. Finally, there is a wide range of applications that prioritize accuracy over speed – including fields like ecosystem conservation and manufacturing and tasks like automated dataset labeling – where RONIN excels.

## 8 CONCLUSION

In this paper, we address the task of object-level out-of-distribution (OOD) detection, aiming to identify cases where an object detector incorrectly detects an OOD object belonging to its known categories. We propose RONIN, a novel method that utilizes pre-trained text-to-image generative models for this task without the need to learn from dedicated in-distribution data. Our experiments show that RONIN consistently attains strong and competitive performance across diverse settings, with promising potential for extension to adjacent tasks such as misclassified in-distribution identification.

ETHICS STATEMENT

This research does not involve human subjects, nor does it introduce any ethical concerns related to privacy, security, or fairness. The methodologies and data used adhere to ethical standards, ensuring no conflicts of interest, harmful applications, or biases. All necessary legal and research integrity guidelines have been followed throughout the study.

REPRODUCIBILITY STATEMENT

All the experiments and ablation studies are fully reproducible. Key assumptions and hyperparameters are outlined in the appendix, and additional information is available in the supplementary materials to ensure clarity and replicability of the results.

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

## A  DATA PROCESSING

Following Du et al. (2022b;a), we used PascalVOC (Everingham et al., 2010) with 20 categories[1] and BDD100k (Yu et al., 2020) with 10 categories[2]. We utilized two subsets of MS-COCO (Lin et al., 2014) and OpenImages (Kuznetsova et al., 2020) as the OOD sets, where all images containing ID categories are removed to ensure the disjoint-category requirement. Additionally, for some British labels such as "Aeroplane" or "Couch", we replaced them with the US counterparts such as "Airplane" or "Sofa" to optimize the usage of CLIP visual-textual representations.

Since our method is zero-shot and does not require training, all the experiments are conducted in subsets taken from the testing set of VOC, BDD, COCO, and OpenImages. To ensure intensive, fairness, and effective evaluations, as well as the balance between ID and OOD objects, we sample a subset of 200 images for BDD-100k testing set and 400 images for PascalVOC, MS-COCO, and OpenImages, as BDD-100k often contain a significantly larger amount of objects per image.

---

[1] PascalVOC contains the following ID labels: Person, Car, Bicycle, Boat, Bus, Motorbike, Train, Aeroplane, Chair, Bottle, Dining Table, Potted Plant, TV Monitor, Couch, Bird, Cat, Cow, Dog, Horse, Sheep.

[2] BDD100k contains the following ID labels: Pedestrian, Rider, Car, Truck, Bus, Train, Motorcycle, Bicycle, Traffic Light, Traffic Sign

## B  IMPLEMENTATION DETAILS FOR BASELINES

For the discriminative zero-shot approaches, MCM (Ming et al., 2022) is originally developed for image-level OOD detection; we adapt it to the object-level setting by applying it over the cropped detections. ODIN (Liang et al., 2018) and Energy score (Liu et al., 2020) are originally developed for non-zero-shot classifier-based settings; we adapt them to the zero-shot object-level setting by using CLIP to classify the detection crops. For alternative generative approaches, we first generate synthetic objects using Stable Diffusion 2, and then extract features with SimCLRv2 (Chen et al., 2020) and fit OOD detectors using standard methods like Mahalanobis (Xiao et al., 2021) and KNN (Sun et al., 2022), for a fair comparison. For SIREN (Du et al., 2022a) and VOS (Du et al., 2022b), we follow their original settings, using the features from the detectors trained with their proposed schemes.

## C  RONIN PERFORMANCE ON CNN-BASED MODEL

Table 6: Object-level OOD detection results of RONIN and comparison with other baselines, with Faster R-CNN as object predictions. The ID dataset is PascalVOC.

| mAP (ID) | Method | MS-COCO FPR@95 ($\downarrow$) / AUROC ($\uparrow$) | OpenImages FPR@95 ($\downarrow$) / AUROC ($\uparrow$) |
|---|---|---|---|
| | ODIN | 54.14 / 85.53 | 47.74 / 88.08 |
| | MCM | 67.15 / 84.54 | 70.16 / 83.43 |
| | Energy Score | 34.69 / 88.67 | 27.42 / 91.89 |
| | Mahalanobis | 66.89 / 78.22 | 50.48 / 83.81 |
| 77.90 | KNN | 54.01 / 85.40 | 46.29 / 84.49 |
| | VOS | 48.15 / 88.15 | 53.47 / 85.64 |
| | SIREN (KNN) | 48.88 / 89.34 | 55.16 / 87.25 |
| | **RONIN** | 32.19 / 90.38 | 22.58 / 93.31 |

Table 6 further present RONIN performances compared to the baseline methods on PascalVOC as ID and MS-COCO/OpenImages at OOD settings, with Faster R-CNN as object detector. Similar to Deformable DETR (Table 1), RONIN still achieved consistent performances, with superior results compared to other baselines. Unlike existing methods with performances constrained by specific architectures (Du et al., 2022a), RONIN provides a strong generality across object detection models, solely dependent on the prediction outcomes without any training effort. Therefore, our proposed methods are highly efficient in modification in the most flexible ways.

## D  INPAINTING PERFORMANCES WITH DIFFERENT MASKING RATIOS

Figure 7 illustrates the performances of RONIN with different ratios between the foreground mask and the semantic background within the predicted box, with the foreground mask covered at 25%, 56%, 81%, and 100% of the bounding box respectively, leaving the remained semantic background ranging from 75% to 0%. As presented in Section 4.1 the foreground $m$ determines the region where the diffusion model DDPM performs inpainting, and the background $s$ provides contextual information to guide the process consistently. On the one hand, a very small region of $m$ prevents DDPM from synthesizing objects globally, whereas a redundant $s$ provides so much information to result in very similar inpainted objects compared to the original ones, both visually and semantically. This led to high similarity scores on both ID and OOD objects, ineffectively separating them. On the other hand, very high $m$ allow DDPM to freely synthesize any type of object conditioned by the label, without semantic information from $s$ to guide the outcome appearance. As a result, the synthesized objects are highly dissimilar to their original counterparts, both in ID and OOD, leading to poor similarity scores. With a reasonable masked region of 81%, equivalent to the side ratio of $r = 0.9$, RONIN both attained the similar shape and appearance of the original objects for high scores in ID, but synthesized objects differently based on predicted labels for low similarity scores on OOD, thus effectively recognizing OOD objects.

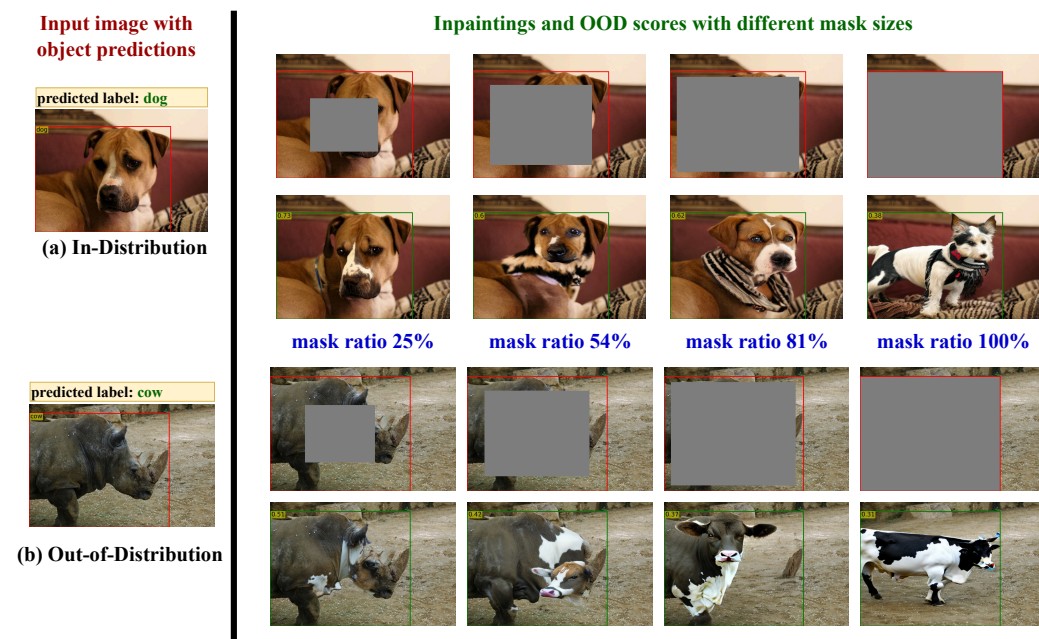

Figure 7: **Contextually inpainting with different masking ratios**. A small masking ratio results in similar inpainting objects compared to their original counterparts, leading to high similarities in both ID and OOD cases. In contrast, a high masking ratio leads to low similarities in both cases. With a reasonable masking ratio of 81%, RONIN maintains the same structure and appearance in the ID case as the original object but renders a different appearance in OOD, hence distinguishing them.

# E    ADDITIONAL RESULTS ON CLASS-WISE VS. OBJECT-WISE INPAINTING

Table 7 provides detailed comparison between the speed of class-wise vs. object-wise inpainting. The two strategies have comparable performance, but object-wise inpainting is slower. Therefore, class-wise inpainting is generally preferred.

Table 7: Runtime performance (second) between class-wise and object-wise inpainting per object.

| ID - OOD setting | Mean predictions / image | | Mean runtime / object | | Speed Up |
|---|---|---|---|---|---|
| | Objects | Labels | Object-wise | Class-wise | (↑) |
| VOC - COCO | 1.82 | 1.21 | 0.85s | 0.59s | ×1.44 |
| VOC - OpenImages | 1.74 | 1.19 | 0.85s | 0.60s | ×1.41 |
| BDD - COCO | 7.41 | 1.92 | 0.85s | 0.27s | ×3.15 |
| BDD - OpenImages | 7.26 | 1.91 | 0.85s | 0.26s | ×3.26 |

# F    ADDITIONAL VISUALIZATION RESULTS

We provide additional visualization of RONIN in different settings. The results are shown in Figure 8 for BDD-100k as ID set and Figure 9 for PascalVOC as ID set.

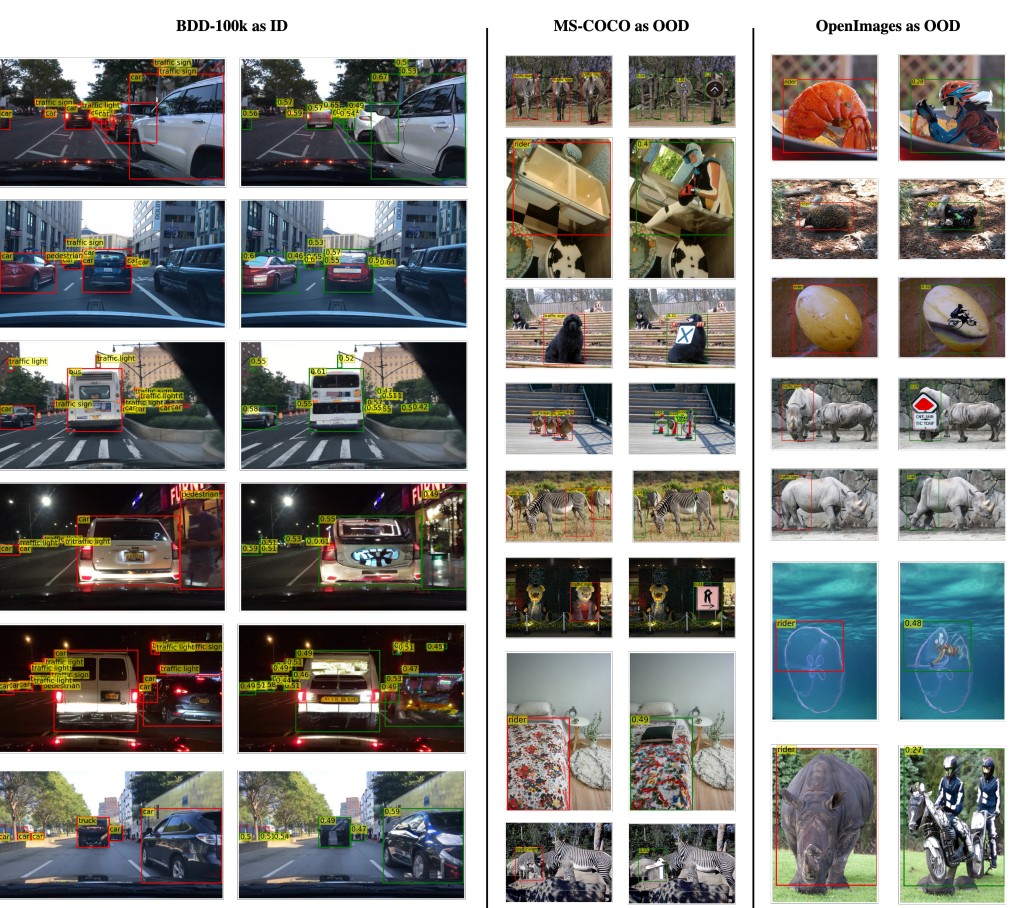

Figure 8: Additional visualization, with BDD-100k as in-distribution set.

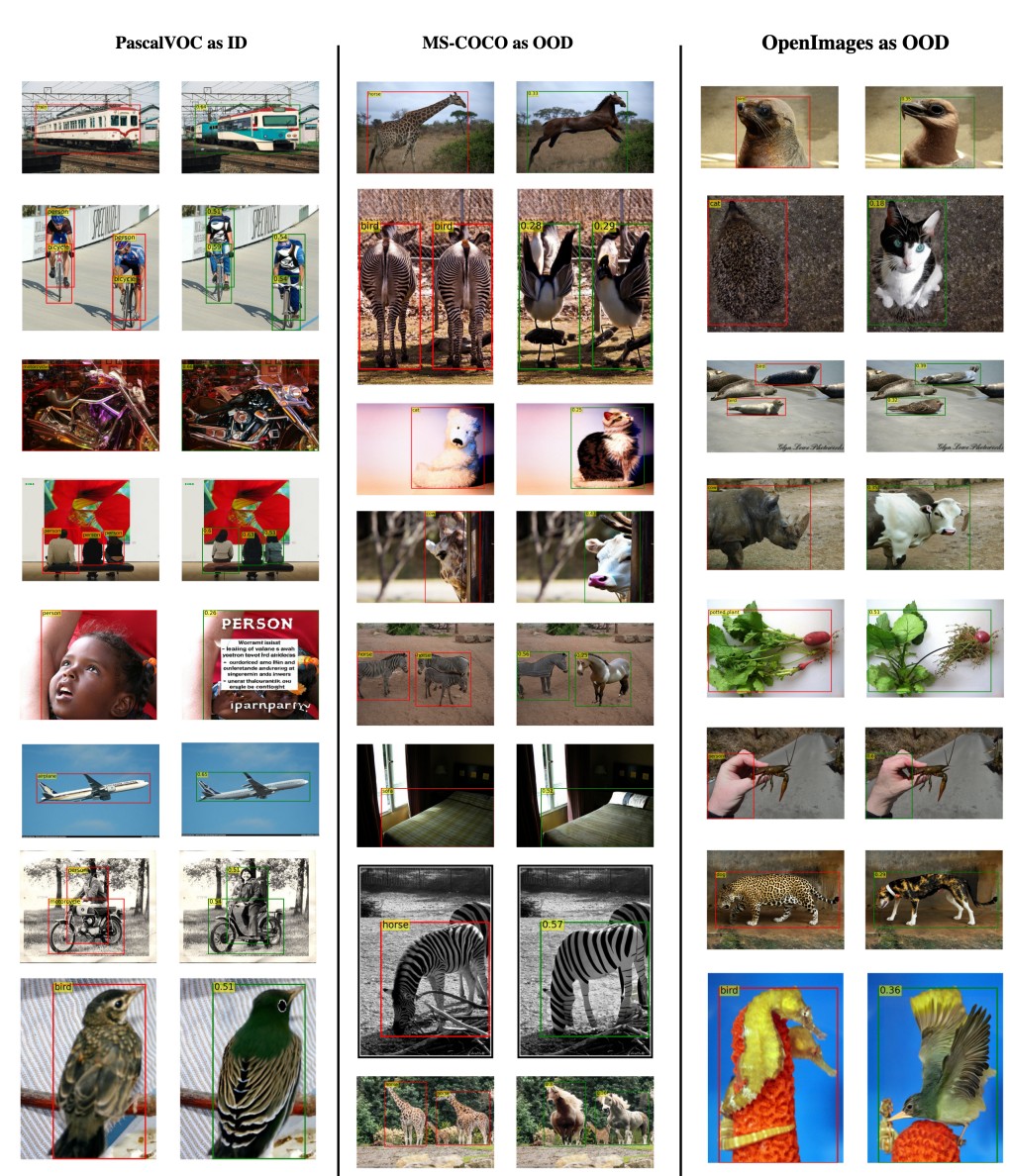

Figure 9: Additional visualization, with Pascal-VOC as in-distribution set.

