# OpenReview forum: "Zero-shot Object-level Out-of-distribution Detection with Context-aware Inpainting"
_ICLR.cc/2025/Conference — Submitted to ICLR 2025_

### Official Review · Reviewer_h9zL · 2024-10-27

**Soundness:** 2
**Presentation:** 3
**Contribution:** 2
**Rating:** 6
**Confidence:** 3

**Summary:**

This paper addresses zero-shot out-of-distribution (OOD) detection in object detection. Since object detection is often overconfident, the OOD scores computed using only the confidence values of object detection are often unreliable.

The proposed method uses text-based image inpainting with Stable Diffusion; The proposed method first input the original image in which the regions of the detected bounding box are masked and the name fo the detected object class to Stable Diffusion to get the synthesized image in which the masked regions are inpainted. OOD detection is performed based on the similarity of the embeddings between the inpainted and the original images.

Experiments using PascalVOC and BDD100k as ID datasets and MS-COCO and OpenImages as OOD datasets show the superiority of the proposed method over several existing OOD detection methods.

**Strengths:**

- The idea of detecting OODs by the similarity of the images before and after inpainting is general, simple and interesting.

- The method of achieving inpainting that preserves the original context associated with the OOD classes by masking the areas that are r% smaller than the bounding box is also simple and interesting.

- The experiments and analysis are generally comprehensive, although there are some serious shortcomings, as discussed below.Although somewhat artificial, the analysis to improve misclassification of ID classes is also interesting.

- The paper is generally well organized and easy to follow.

**Weaknesses:**

**W1.** The idea of using image inpainting for OOD detection is novel in itself, but its technical novelty is somewhat limited because it is basically a combination of existing pre-trained image inpainting and classifiers.


**W2.** OOD detection using image inpainting, while interesting, yields several weaknesses.

*W2-1.* As the authors themselves discuss, high-quality inpainting is computationally time-consuming, as inference also takes a long time for each image. This can be a drawback since object detection has many applications where processing time is crucial.

*W2-2.* If the ID class and OOD class are close, image inpainting with clear class distinctions and precise similarity evaluation may become difficult, resulting in less accurate OOD detection. It would be good to evaluate detection performance on datasets that include fine-grained categories and attributes, for example, LVIS and FG-OVD. This would also be relevant to the discussion around Line 076. It would also be good to discuss the object categories that are difficult to inpaint and their impact on performance.

*W2-3.* The results of inpainting are dependent on the initial noise, which can lead to large variations in OOD detection performance. Discussion on this point would be appreciated.


**W3.** Comparisons with existing methods are somewhat lacking. In this paper, the problem is defined as the task of binary classification of each detected object bbox as to whether it is ID or OOD. According to the task definition, a simple baseline would be to apply a zero-shot OOD detection method to each bbox. In particular, the proposed method uses CLIP, but there are other CLIP-based zero-shot OOD detection methods (e.g., CLIPN [Wang et al., ICCV'23]) and few-shot OOD detection methods (e.g., LoCoOp [Miyai et al., NeurIPS'23] and NegPrompt [Li et al., CVPR'24]). Comparisons with these methods should be included.


**W4.** The analysis presented in Table 4 is not exhaustive and does not fully support the validity of the triplet similarity in Eq. 4. In general, adding more hyperparameters should help improve accuracy. To demonstrate the validity of Equation 4, it is necessary to show exhaustive results for sensitivity to alpha and beta, with similarity(ori, \hat{y}) and similarity(ori, inp) each showing performance improvement over a wide range of alpha and beta.


**W5.** Modern object detectors are often open-vocabulary. Demonstrating that the proposed method is effective even in open-vocabulary scenarios is preferable.


**W6.** Other minor points

- RONIN (the name of the proposed method) first appears in the caption of Fig. 1 and Line 059 in the introduction section, but without saying what it stands for. There should be the definition.

- Bold b's appear in Lines 119 and 120 should be italic.

- Fig. 1 and Fig. 2 have considerable overlap in information and could be combined into one.

**Questions:**

Since the use of image inpainting is the most important idea of the proposed method, its strengths and weaknesses should be fully discussed. In this regard, I would expect thorough responses on **W2**. I would also like to see answers to the lack of comparison with existing methods (**W3**) and the lack of analysis (**W4**), as these are critical issues to clarify the validity and effectiveness of the proposed method. Given the recent emphasis on open vocabulary scenarios for object detection, I would expect an answer for **W5** as well.

---

> ### Author Response · Authors · 2024-11-22
> **Response to Reviewer h9zL (1/2)**
>
> Thank you for your detailed and constructive feedback. Below, we respond to most, if not all, of your comments at this stage. We apologize in advance if the response is long.
>
> ## W1: Limited in technical novelty
> Thank you for recognizing the novelty of our framework for OOD detection based on image inpainting. We agree that, having a more sophisticated or dedicated integration of inpainting and classification models together would make our method appear even more novel and appealing. However, we kindly think that, by doing so, inadvertently degrades the generality of our framework. Overall, our goal is to develop a zero-shot approach where newly released conditional generative models and object detectors can be easily plug-and-play without any extra modification efforts. We respectfully believe keeping our framework simple is actually a strength, not a weakness.
>
> ## W2: Potential weaknesses using inpainting
>
> We would like to discuss the weaknesses as follows:
>
> #### **W2.1. Computation Time**
>
> Thank you for raising this concern. We would like to note that it is often possible to construct shorter diffusion schedules that maintain comparable performance. Also, RONIN does not need extremely high-quality inpaintings, but rather good enough ones that make ID and OOD separable. Below we compare RONIN's performance using the original 20-step inpainting schedule vs. an optimized 2-step schedule. The 2-step schedule preserves most of the performance while running in near real-time. Finally, we respectfully emphasize that object detection is crucial in both online and offline systems, and certain application tasks favor accuracy over time performances. We will further clarify and elaborate on the time and resource computational in the final version of our paper.
>
> **OOD Detection Performance**
>
> |                  |  RONIN with 20 steps (original schedule)    | RONIN with 2 steps (improved schedule)            |
> |-------------------|----------------|----------------|
> |                   | FPR@95 / AuROC | FPR@95 / AuROC |
> | VOC - COCO |    27.42 / 91.09  |    37.11 / 87.54  |
> | VOCO - OpenImages | 18.04 / 93.34  | 28.48 / 90.52  |
> | BDD - COCO        | 30.16 / 92.77  | 28.57 / 92.98  |
> | BDD - OpenImages  | 30.00 / 91.60  | 31.11 / 91.37  |
>
> **Runtime in Seconds per Object**
>
> |ID-OOD setting|RONIN with 20 steps |RONIN with 2 steps |
> |-|-|-|
> |VOC-COCO|0.35|0.034|
> |VOC-OpenImages|0.42|0.041|
> |BDD-COCO|0.19|0.018|
> |BDD-OpenImages|0.18|0.017|
>
> #### **W2.2. Near-OOD performance.**
> Thank you for pointing out this. We recognize that near-OOD is indeed a very challenging case for the standard OOD detection algorithms. We indeed conducted a study about near-OOD in the appendix of our original paper, in particular, Appendix F (Line 820 - 858) of the original manuscript. What we found out is that using off-the-shelf generative models can actually have the strength to do near-OOD detection. Let’s say we have an implicated input as a photo consisting of OOD objects *“zebra”*, incorrectly predicted as an ID label *“horse”*. Because the two classes are really similar, if we just inpaint the zebra to become a horse, it may result in something quite similar to the original zebra, making near-OOD detection ineffective. Rather than that, we can pre-anticipate several near-OOD classes of the ID label. So, even though the ID class is a horse, we have already roughly guessed what are the near-OOD classes, including zebra. Then, based on that, we deliberately condition the diffusion model to generate a horse, but it doesn't look like any near-OOD class-like objects, including zebra. This will make our inpainted results even more aligned with the predicted ID label, and by doing so enhance the way our framework is robust to the near-OOD.
>
> Based on your suggestion, we are investigating further near-OOD detection by extending what we conducted in our original appendix. In particular, we specifically look into some image samples that contain certain objects strongly similar to the ID labels in the VOC-COCO setting, stimulating the near-OOD samples to the PascalVOC object detector. We recognize that across these samples, our framework RONIN is able to tackle most of these cases effectively for near-OOD detection. We ensure that we will clarify this in detail in the main paper.
>
> #### **W2.3. Noise initialization for inpainting.**
> Thank you for your comment. We perform RONIN by giving 5 random starting noises and computing overall OOD measurements FPR@95 and AuROC across all samples. The small standard derivation suggests that RONIN does not suffer considerably under the noising of the diffusion model.
>
> |                  | FPR@95         | AuROC         |
> |------------------|----------------|---------------|
> | VOC - COCO       | 28.12% ± 1.38  | 91.12% ± 0.24 |
> | VOC - OpenImages | 12.61% ± 1.04  | 92.99% ± 0.13 |
> | BDD - COCO       | 29.90% ± 0.62  | 92.23% ± 0.29 |
> | BDD - OpenImages | 32.22% ± 3.06  | 90.40% ± 0.81 |

---

> ### Author Response · Authors · 2024-11-22
> **Response to Reviewer h9zL (2/2)**
>
> ## W3: Comparison with additional baselines
> Thank you for your suggestion. Below we provide additional comparison with the most recent CLIPN [1]. The current results suggest that our framework, RONIN, still achieves superior performance.
>
> |                   | CLIPN          | RONIN          |
> |-------------------|----------------|----------------|
> |                   | FPR@95 / AuROC | FPR@95 / AuROC |
> | VOC - COCO        | 43.09 / 85.45  | 27.42 / 91.09  |
> | VOCO - OpenImages | 41.74 / 89.31  | 18.04 / 93.34  |
> | BDD - COCO        | 28.89 / 92.14  | 30.16 / 92.77  |
> | BDD - OpenImages  | 44.76 / 85.78  | 30.00 / 91.60  |
>
>
> ## W4: Clarification on the ablation study of similarities (Table 4)
> Thanks for the constructive comment. We acknowledge that Table 4 could be improved to be more organized and supportive of our algorithm design (Equation 4). We will revise it in the final version and here is our plan.
>
> In this new table, we will intergrade two similar ablation studies to support our design for the triplet similarity. In the first study, we prove that having all three types of similarity is important and necessary for effective similarity measurements. In the second study, given the triplet score with all three similarities, we empirically prove that focusing more on the vision-language similarity rather than the visual similarity, i.e. $\alpha > \beta$, supports our method RONIN to perform at its peak.
>
> Additionally, we recognized that the performance of our framework RONIN can be sensitive given different choices of both $\alpha$ and $\beta$, making them effective hyperparameters if we want to tune our methods to tackle difficult settings. Initially, increasing $\alpha$ and $\beta$ does improve the accuracy, as you pointed out. However, our experiments suggest that keeping $\beta = 1$ is actually the most stable choice, and if having $\alpha \geq 4$, the performances of our framework then do not make significant improvements anymore. For convenience and simplicity, we just simply select $\beta = 1$ and $\alpha = 2$ for our algorithm design as in Equation 4.
>
> ## W5: The effectiveness of RONIN in open-vocabulary scenarios
> Thank you for raising this very interesting question. To our understanding, open-vocabulary means the object detector have the ability to detect any kind of object if someone provides a set of which objects will be detected. In other words, it requires the user to provide a set of candidate ID classes to let it operate. So based on this understanding, we see no blockage of applying our methods for open-vocabulary object detection. Here we conducted a very simple study by taking several image data from the VOC-COCO setting, containing horse (in-vocabulary) and zebra (out-of-vocabulary), and we ran GroundingDINO [2], a state-of-the-art open-vocabulary object detection. The object detector easily detects horses, but we found out there are a few false-detected zebras as horses. These false-detected can be identified with our framework, RONIN, which is particularly discussed in our response to W.2.2.
>
> ## W6: Minor errors
> Thank you for pointing out that, and we will correct these errors in the final revision of our paper. We would like to explain that, the name of our framework **RONIN** is the abbreviation of  _"Ze**R**o-shot OOD C**ON**textual **IN**painting"_.
>
> Again, we appreciate your valuable and constructive feedback, which will significantly improve our study. We kindly hope that you find these responses satisfying and give us a chance to clarify any remaining unclear points.
>
> [1] Wang et. al. "Clipn for zero-shot ood detection: Teaching clip to say no." ICCV 2023.
> [2] Liu et. al. "Grounding DINO: Marrying DINO with Grounded Pre-Training for Open-Set Object Detection." ECCV 2024.

---

> > ### Comment · Reviewer_h9zL · 2024-11-25
> >
> > I would like to thank the authors for providing responses to my questions. I found that most of the reviewers had similar concerns, including lack of technical novelty, concern about inference time, and lack of comparative experiments and analysis. I think the authors' responses addressed these questions partially, but not completely. More specifically,
> >
> > **W1**. I am satisfied with the authors' comment. Although I raised the lack of algorithmic novelty as a weakness, I would not put much emphasis on this point because I agree with the simplicity and interest of the idea, as I mentioned in the Strengths section.
> >
> > **W2**. After reading the authors' response, I think the computation time of the proposed method is a clear weakness; as Reviewer os8e also said, this cannot be ruled out due to the lack of comparison of inference times.
> > In addition, the few-round version causes a clear drop in accuracy, and it cannot be determined at this stage that the proposed method achieves a good trade-off.
> >
> > **W3**. The authors have provided additional comparative experiments with one of the methods I (and Reviewer oHFd) have mentioned. However, the comparisons with the other three methods remain unknown.
> >
> > Regarding **W4** and **W5**, I cannot make any comments at this stage because no clear justification has been provided.
> >
> > As above, the rebuttal has not been very convincing so far. But I still feel that the idea of detecting OOD by the similarity of images before and after inpainting is interesting, so I would retain my original rating.

---

### Official Review · Reviewer_os8e · 2024-10-31

**Soundness:** 3
**Presentation:** 3
**Contribution:** 2
**Rating:** 5
**Confidence:** 4

**Summary:**

This paper proposes a new object-level OOD detection method called RONIN, which leverages the off-the-shelf diffusion model to replace detected objects with inpainting.  RONIN adopts a context-aware class-wise inpainting method and combinations of similarity assessment methods. Experimental results show that the proposed method outperforms existing methods in three of 4 settings on Pascal-VOC and BDD-100k.

**Strengths:**

- S1. Using diffusion inpainting strategy for object-level OOD detection is novel.

- S2. The proposed method outperforms existing methods in three of four settings.

- S3. This paper is well written.

**Weaknesses:**

- W1. The inference times of comparison methods are not reported. I am concerned that this method may have a longer inference time than the comparison methods. Since the actual application of object-level OOD detection is in areas that require real-time inference, such as autonomous driving, I believe it is necessary to accurately report the inference time of comparison methods.

- W2. I wonder the reason why RONIN’s performance is lower than the comparison methods in BDD-100k vs. OpenImages. It might be better to include the reason for this result. This is because BDD-100k is a driving dataset and closer to the real-world application than VOC.

**Questions:**

I would like to know the inference time of the comparison methods.

I also wonder the reason why RONIN’s performance is lower than the comparison methods in BDD-100k vs. OpenImages.


I am willing to raise the score, taking into account discussions with the author and the opinions of other reviewers.

---

> ### Author Response · Authors · 2024-11-22
> **Response to Reviewer os8e**
>
> We thank your valuable feedback to improve our manuscript further. Below, we respond to most, if not all, of your comments at this stage.
>
> ## W1: Inference time comparison
> Thank you for your suggestion. We have just come up with a new diffusion inpainting schedule, allowing us to perform our framework RONIN significantly faster by 10 times with only 2 denoising steps, while still maintaining an approximate performance compared to our official results in the manuscript. Despite the diffusion model can be costly, this new schedule allows RONIN to perform in near real-time. Furthermore, as discussed in Lines 042-048 and 474-476 of the original manuscript, we believe RONIN can excel in both online and offline object detection tasks, especially in scenarios where accuracy performance is favored over speed. Finally, with the simple and generalized framework, we believe newly released one-step diffusion models with real-time performance can be easily plug-and-play into our framework to make it more effective and efficient. We believe eventually RONIN can perform on par with other baselines in terms of inference time.
>
> **Runetime in Seconds per Object**
>
> |ID-OOD setting|RONIN with 20 steps (original schedule)|RONIN with 2 steps (improved schedule)|
> |-|-|-|
> |VOC-COCO|0.35|0.034|
> |VOC-OpenImages|0.42|0.041|
> |BDD-COCO|0.19|0.018|
> |BDD-OpenImages|0.18|0.017|
>
> **OOD Detection Performance**
>
> |                  | RONIN (20 steps)          | RONIN (2 steps)          |
> |-------------------|----------------|----------------|
> |                   | FPR@95 / AuROC | FPR@95 / AuROC |
> | VOC - COCO       | 27.42 / 91.09  | 37.11 / 87.54  |
> | VOCO - OpenImages | 18.04 / 93.34  | 28.48 / 90.52  |
> | BDD - COCO        | 30.16 / 92.77  | 28.57 / 92.98  |
> | BDD - OpenImages  | 30.00 / 91.60  | 31.11 / 91.37  |
>
>
> ## W2: Performance clarification on BDD vs OpenImages
> Thank you for your pointing out that, and we would like to clarify the performance as follows. In the main Table 1, across the 4 OOD settings, our proposed framework RONIN is the only method that performs competitively and consistently, while the other baselines fluctuate quite a bit. On the BDD vs. OpenImages setting, RONIN achieves 30.00 FPR@95TPR, and 91.60 AuROC. Although this is not a state-of-the-art performance, these numbers are still strong enough to allow RONIN to be among the top methods. On the other hand, BDD and OpenImages datasets are two significantly distinct dataset distributions. Under very effective visual embedding, such as SimCLR, metric-based learning methods like KNN or Mahalanobis perform extremely robustly to achieve peak performance. Despite that, they still underperformed RONIN in other scenarios.
>
> Again, we appreciate your valuable feedback on improving our study. Please also kindly give us a chance to clarify any remaining unclear points.

---

> > ### Comment · Reviewer_os8e · 2024-11-24
> > **Response to Authors' Rebuttal**
> >
> > Thank you for addressing my feedback. However, I would like to raise the following concerns regarding the revisions:
> >
> > - In W1, I requested the reporting of inference times for comparison methods. However, the authors only reported the inference time of their proposed method and did not include the inference times for comparison methods. Therefore, this concern has not been addressed.
> > - The authors claim that faster diffusion models may be proposed in the future, which will resolve the efficiency issue. However, there is a trade-off between performance and speed, so it is unclear whether RONIN's performance is sufficient.  Additionally, to demonstrate the model's scalability, it is necessary to actually verify it with a wider variety of diffusion models, but this point has not been sufficiently discussed in the paper.
> > - The 2-step diffusion model shows significantly lower performance on VOC dataset, falling behind other comparative methods. This suggests the improvement in performance may be limited.
> > - While they claim the method is effective in scenarios where accuracy performance is favored over speed, there are no experiments with such datasets. In the paper, examples like ecosystem conservation and manufacturing are mentioned at L475, and I believe conducting experiments with such datasets would make their claims more convincing. However, since they only evaluate performance on common benchmarks like PASCAL-VOC and BDD100K, I remain skeptical about applications in these areas.

---

### Official Review · Reviewer_9VVi · 2024-11-02

**Soundness:** 3
**Presentation:** 2
**Contribution:** 2
**Rating:** 5
**Confidence:** 3

**Summary:**

This paper introduces a zero-shot object-level out-of-distribution detection method using context-aware inpainting to improve the model's ability to identify unseen objects. By leveraging generative models for class-conditioned inpainting, the differences between detected and original objects serve as indicators of recognition errors. Experiments demonstrate that this approach outperforms existing zero-shot and non-zero-shot OOD detection methods.

**Strengths:**

1. Proposes a novel approach by integrating generative models for OOD detection, offering a solution distinct from traditional methods.
2.  Supported by data, showing exceptional performance across multiple benchmark datasets.
3. The method does not require modifications to pre-trained models, facilitating integration into existing systems.

**Weaknesses:**

1. High reliance on generative models may be limiting, especially if the models are inaccurate or under-trained.
2. The need for extensive image generation and comparison could lead to high resource consumption.
3. Further validation on a variety of datasets is needed to ensure broad applicability and robustness.

**Questions:**

Please refer to the Weaknesses box.

---

> ### Author Response · Authors · 2024-11-22
> **Response to Reviewer 9VVi**
>
> Thanks for your valuable feedback to improve our manuscript further. Below, we respond to most, if not all, of your comments at this stage.
>
> ## W1: High reliance on the generative model
> Thank you for your comment, and we agree that our method, RONIN, does rely on an effective off-the-shelf diffusion model with well-trained, generalized performances for doing OOD detection in a zero-shot manner. This reliance is actually similar to several existing methods, including [1], [2], or [3] that also develop their own methods based on effective pre-trained generative models. We deliberately tried to keep the inpainting model intact to ensure the generalizability of our framework, as newly released off-the-shelf generative models can be conveniently plug-and-play without additional effort. However, to demonstrate the effectiveness of our framework without heavy reliance on the performance of the generative model, we conducted an additional experiment where we compared the default StableDiffusion 2 with two other sub-optimal generative models, StableDiffusion 1 and StableDiffusionXL, for our framework. Additionally, we further blurred the generated outcomes to stimulate an “underperformed” diffusion model with poor inpainting results. The result suggests that our framework is still able to perform similarly without significant derivation. In fact, we found that RONIN can be effectively operated with an acceptable reconstruction, without the the need for high-quality or detailed generated outcomes.
>
> | **RONIN with different Diffusion Models** | **VOC vs COCO** | **VOC vs Open images** |
> |:-----------------------------------------:|:---------------:|:---------------------:|
> |                                           |   FPR / AuROC   |      FPR / AuROC      |
> | StableDiffusion 1                         |   29.9 / 89.36  |     23.70 / 92.36     |
> | StableDiffusion 1 + blur                  |  33.81 / 88.52  |     25.22 / 91.66     |
> | StableDiffusion XL                        |  32.99 / 90.83  |     22.17 / 92.61     |
> | StableDiffusion XL + blur                 |  31.96 / 90.64  |     21.96 / 92.09     |
> | StableDiffusion 2                         |  27.42 / 91.41  |     18.91 / 92.96     |
> | StableDiffusion 2 + blur                  |  27.42 / 91.22  |     19.35 / 92.67     |
>
> [1] Zhao et. al. "Unleashing Text-to-Image Diffusion Models for Visual Perception." CVPR 2024.
>
> [2] Luo et. al. "Diff-Instruct: A Universal Approach for Transferring Knowledge From Pre-trained Diffusion Models." NeurIPS 2024.
>
> [3] Mou et. al. "T2I-Adapter: Learning Adapters to Dig out More Controllable Ability for Text-to-Image Diffusion Models." AAAI 2024
>
> ## W2: Clarification on the high resource consumption
>
> Thank you for raising the concern. We find out that operating our framework with both the image inpainting and similarity comparison is actually not costly, especially given the modern computational resources. As discussed in Session 4.2 (Line 191-211) of the original manuscript, we designed a “class-wise inpainting” strategy to significantly reduce the number of inpainting per image, making RONIN highly efficient. Additionally, for the rebuttal, we have just designed a new inpainting schedule and denoising process that supports the generative models to perform image inpainting with only two steps but still approximates maintaining performance compared with our reported performance in the original paper, which resulted from 20 steps of inpainting. This means the new schedule allows RONIN to perform 10 times faster, approaching near real-time performance. Below we provide the time comparison of our framework RONIN, both with the original schedule of 20 steps and the improved schedule of 2 steps across the 4 main settings. We kindly ensure we will elaborate on this finding in the final version of our paper.
>
> **Runtime in Seconds per Object**
>
> |ID-OOD setting|RONIN with 20 steps (original schedule)|RONIN with 2 steps (improved schedule)|
> |-|-|-|
> |VOC-COCO|0.35|0.034|
> |VOC-OpenImages|0.42|0.041|
> |BDD-COCO|0.19|0.018|
> |BDD-OpenImages|0.18|0.017|
>
> **OOD Detection Performance**
>
> |                  | RONIN (20 steps)          | RONIN (2 steps)          |
> |-------------------|----------------|----------------|
> |                   | FPR@95 / AuROC | FPR@95 / AuROC |
> | VOC - COCO       | 27.42 / 91.09  | 37.11 / 87.54  |
> | VOCO - OpenImages | 18.04 / 93.34  | 28.48 / 90.52  |
> | BDD - COCO        | 30.16 / 92.77  | 28.57 / 92.98  |
> | BDD - OpenImages  | 30.00 / 91.60  | 31.11 / 91.37  |
>
>
> ## W3: Further validation
> Thank you for your suggestion. We are actively developing new OOD settings and validations, as preparing the dataset is time-consuming. We will update you in the next few days.
>
> Again, we appreciate your valuable opinions and reviews, which will significantly improve our study. Please also kindly give us a chance to clarify any remaining unclear points.

---

> > ### Comment · Reviewer_9VVi · 2024-11-27
> >
> > After checking the responses to my questions, I believe that most of my concerns have been resolved. Considering the originality and quality of this article, I decided to maintain my rating as “5: marginally below the acceptance threshold.”

---

### Official Review · Reviewer_oHFd · 2024-11-02

**Soundness:** 2
**Presentation:** 2
**Contribution:** 2
**Rating:** 5
**Confidence:** 3

**Summary:**

This paper presents RONIN for object-level out-of-distribution (OOD) detection. Given a detected object box and its predicted class label, RONIN first performs class-conditioned inpainting on the image with the object box removed. It then compares the similarities between the original detected objects and their inpainted versions to identify OOD instances.

**Strengths:**

1. The concept of using inpainting and comparing similarity scores for OOD detection is both reasonable and straightforward.
2. The proposed method outperforms zero-shot baseline methods by noticeable margins.

**Weaknesses:**

1. The technical contribution is limited. The proposed method simply combines image inpainting with similarity comparison, both of which are straightforward and well-established techniques. Overall, the original technical contribution is insufficient.
2. The paper lacks comparisons with recent methods. The latest method cited in Table 1 is from 2022. Comparisons with more recent approaches, such as [1] and [2], are expected.
3. It seems from Table 4 the results are highly sensitive to the choice of $\alpha$ and $\beta$, suggesting that careful parameter fine-tuning may be required.

[1] Li et al. "Learning Transferable Negative Prompts for Out-of-Distribution Detection." CVPR 24

[2] Wang et al. "CLIPN for Zero-Shot OOD Detection: Teaching CLIP to Say No." ICCV 23

**Questions:**

1. The authors did not conduct experiments on ImageNet-1K, a commonly used dataset for out-of-distribution (OOD) detection. Could the authors provide results on this dataset as well?
2. Instead of performing image inpainting, would generating a complete image of the predicted category using a pre-trained diffusion model work? Can the authors include comparisons with this simple baseline?
3. In Line 261, it is mentioned that "one can use a thresholding method to identify OOD objects effectively". What is the threshold used in the experiments?

---

> ### Author Response · Authors · 2024-11-22
>
> Thank you for your constructive feedback. Please find our responses below:
>
> ## W1: Technical contribution
> We respectfully believe that our method is actually not limited and insufficient. We would like to emphasize that our work proposes a *novel OOD detection framework* based on off-the-shelf generative models. Our framework is intentionally designed to be modular, keeping the object detector and the off-the-shelf generative model intact, while investigating how to combine them effectively for performing OOD detection in a zero-shot manner. By doing so, the framework can easily generalize to future object detectors and newly released generative models with minimal modification effort. We believe that prioritizing *simplicity* and *generalizability* enhances the impact and applicability of our work, rather than diminishing its contribution.
>
> ## W2: Additional comparison with other baseline methods
> Thank you for your suggestion. Below we present some additional experiments with a newer baseline CLIPN [1] across 4 settings, where our framework RONIN still outperforms the baseline.
>
> |                   | CLIPN          | RONIN          |
> |-------------------|----------------|----------------|
> |                   | FPR@95 / AuROC | FPR@95 / AuROC |
> | VOC - COCO        | 43.09 / 85.45  | 27.42 / 91.09  |
> | VOCO - OpenImages | 41.74 / 89.31  | 18.04 / 93.34  |
> | BDD - COCO        | 28.89 / 92.14  | 30.16 / 92.77  |
> | BDD - OpenImages  | 44.76 / 85.78  | 30.00 / 91.60  |
>
> ## W3: The sensitivity of $\alpha$ and $\beta$ in Table 4
>
> Thank you for your comment. We would like to clarify that performance fluctuations mainly occur when certain similarity measurements are entirely excluded (row 1-2 of Table 4). However, when all similarity measurements are included, the performance is more consistent (rows 3-6 of Table 4). We recognize that varying the values of $\alpha$ and $\beta$ can lead to performance differences. This can in fact be beneficial for having the flexibility for hyperparameter tuning to meet specific practical needs. Our empirical results indicate that $\beta = 1$ and $\alpha = 2$ perform robustly across diverse datasets, making them reliable default choices.
>
> ## Q1. The experiments on ImageNet-1K
> Thank you for your suggestion. We would like to clarify that, ImageNet-1K is commonly used for *image-level* OOD detection (e.g. for image classification tasks), rather than *object-level* tasks, such as object detection. Our paper investigates the object-level OOD problem, and the four datasets we experiment on follow the existing object-level works [2,3].
>
> ## Q2. The simple baseline of generating an image instead of inpainting an image
> Thank you for your suggestion. In fact, our implementation of the KNN baseline is similar to the suggested simple baseline. For KNN, we first construct a dataset by directly generating images of the ID classes using stable diffusion, and at inference time, we compare the detected objects to images in the generated dataset. Results in Table 1 show that RONIN consistently outperforms this approach.
>
> ## Q3. Clarification on the choice of threshold
> Thank you for pointing out this. We would like to clarify that in our evaluation, AuROC is calculated as the area under the curve of TPR vs. FPR at all possible decision thresholds; FPR@95 TPR is calculated at the threshold where the in-domain achieves 95% true positive rate. In practical use, the threshold can be selected based on criteria such as achieving a 95% TPR.
>
>
> [1] Wang et al. “Clipn for zero-shot ood detection: Teaching clip to say no.” ICCV 2023.
>
> [2] Du et al. “VOS: Learning What You Don't Know by Virtual Outlier Synthesis.” ICLR 2022.
>
> [3] Du et al. “SIREN: Shaping Representations for Detecting Out-of-Distribution Objects.” Neurips 2022.

---

> > ### Comment · Reviewer_oHFd · 2024-11-27
> >
> > Thank authors for the rebuttal. My concern regarding the lack of comparisons with recent methods has been addressed.  However, I am still not fully convinced about the technical contribution of the paper, and I share the concerns raised by other reviewers regarding the computational cost, which I consider to be a weakness. As such, I will maintain my original score.

---

### Author Response · Authors · 2024-11-22
**General Response**

We thank all the reviewers for their time and valuable comments. We appreciate that the reviewers found our approach novel (reviewer 9VVi, os8e), our performance strong (reviewer oHFd, 9VVi, os8e), and our paper well-written (reviewer os8e, h9zL). Below, we address some commonly raised concerns:

**[Technical Contribution]** We recognize that there are differing opinions on the novelty of our proposed method. We would like to emphasize that our work proposes a *novel OOD detection framework* based on off-the-shelf generative models. Our framework is intentionally designed to be modular, keeping the object detector and the off-the-shelf generative model intact, while investigating how to combine them effectively for performing OOD detection in a zero-shot manner. By doing so, the framework can easily generalize to future object detectors and newly released generative models with minimal modification effort. We believe that prioritizing *simplicity* and *generalizability* enhances the impact and applicability of our work, rather than diminishing its contribution.

**[Runtime]** We report RONIN’s runtime in seconds per object. Additionally, we note that it is often possible to construct shorter diffusion schedules that maintain comparable performance. To illustrate the speed-performance tradeoff, we also present RONIN’s results using a 2-step inpainting schedule.

**Number of Seconds Per Object Processed**

|ID-OOD setting|RONIN with 20 steps (original schedule)|RONIN with 2 steps (improved schedule)|
|-|-|-|
|VOC-COCO|0.35|0.034|
|VOC-OpenImages|0.42|0.041|
|BDD-COCO|0.19|0.018|
|BDD-OpenImages|0.18|0.017|

**OOD Detection Performance**

|                  | RONIN (20 steps)          | RONIN (2 steps)          |
|-------------------|----------------|----------------|
|                   | FPR@95 / AuROC | FPR@95 / AuROC |
| VOC - COCO       | 27.42 / 91.09  | 37.11 / 87.54  |
| VOCO - OpenImages | 18.04 / 93.34  | 28.48 / 90.52  |
| BDD - COCO        | 30.16 / 92.77  | 28.57 / 92.98  |
| BDD - OpenImages  | 30.00 / 91.60  | 31.11 / 91.37  |


**[Comparison with More Recent Methods]** We thank the reviewers for suggesting comparisons with more recent baselines. Below, we report a comparison with CLIPN [1]. Results show that RONIN consistently outperforms CLIPN.

|                   | CLIPN          | RONIN          |
|-------------------|----------------|----------------|
|                   | FPR@95 / AuROC | FPR@95 / AuROC |
| VOC - COCO        | 43.09 / 85.45  | 27.42 / 91.09  |
| VOCO - OpenImages | 41.74 / 89.31  | 18.04 / 93.34  |
| BDD - COCO        | 28.89 / 92.14  | 30.16 / 92.77  |
| BDD - OpenImages  | 44.76 / 85.78  | 30.00 / 91.60  |

Thanks again for all the insightful comments and suggestions. Please do not hesitate to let us know if there are additional concerns. We will do our best to address them in the remaining rebuttal period.

[1] Wang et al. "Clipn for zero-shot ood detection: Teaching clip to say no." ICCV 2023.

---

### Author Response · Authors · 2024-11-27
**General response on a revised manuscript**

We sincerely thank all reviewers for their thoughtful comments and feedback. Based on their suggestions, we have revised our manuscript to address most, if not all, of their major concerns. The revised sections are highlighted in blue for clarity. Additionally, we will continue to address any remaining concerns promptly during the rebuttal period. If there are any remaining concerns that are not addressed, please let us know.

---

### Meta-Review · Area_Chair_VuDe · 2024-12-16

**Metareview:**

This paper proposes a novel zero-shot out-of-distribution detection method for object detectors based on inpainting object bounding boxes with stable diffusion. If class-conditioned inpainting deviates more from the original this is taken as an indicator of the wrong class being used for conditioning.
The reviewers report several strengths: interesting idea to use inpainting for OOD, improvements over baseline,
Weaknesses include: limited technical novelty, missing comparison to recent methods, sensitivity to hyperparameters, and inference time.

**Additional Comments On Reviewer Discussion:**

In response to the reviews the authors provided a rebuttal as well as an updated version of the manuscript.
After taking into account the rebuttal and other reviews there are 3 (weak) negative recommendations and one (weak) positive recommendation.
While the idea of class-conditional diffusion-based inpainting for OOD detection is interesting, the paper has several weaknesses including limited algorithmic novelty and misses comparisons with several state-of-the-art OOD detection methods, and comparison in terms of run-time. Therefore the AC follows the majority recommendation for this paper.

---

### Decision · Program_Chairs · 2025-01-22

Reject